# Genome Sequencing in an Individual Presenting with 22q11.2 Deletion Syndrome and Juvenile Idiopathic Arthritis

**DOI:** 10.3390/genes15040513

**Published:** 2024-04-19

**Authors:** Ruy Pires de Oliveira-Sobrinho, Simone Appenzeller, Ianne Pessoa Holanda, Júlia Lôndero Heleno, Josep Jorente, Társis Paiva Vieira, Carlos Eduardo Steiner

**Affiliations:** 1Genética Médica e Medicina Genômica, Departamento de Medicina Translacional, Faculdade de Ciências Médicas, Universidade Estadual de Campinas (Unicamp), Campinas 13083-888, SP, Brazil; ruy.pires@uol.com.br (R.P.d.O.-S.); ianne@unicamp.br (I.P.H.); londero@unicamp.br (J.L.H.); jorente@unicamp.br (J.J.); tpvieira@unicamp.br (T.P.V.); 2Departamento de Ortopedia, Reumatologia e Traumatologia, Faculdade de Ciências Médicas, Universidade Estadual de Campinas (Unicamp), Campinas 13083-888, SP, Brazil; appenzel@unicamp.br

**Keywords:** 22q11.2 deletion syndrome, DiGeorge syndrome, juvenile rheumatoid arthritis, juvenile idiopathic arthritis, Whole Genome Sequencing

## Abstract

Juvenile idiopathic arthritis is a heterogeneous group of diseases characterized by arthritis with poorly known causes, including monogenic disorders and multifactorial etiology. 22q11.2 proximal deletion syndrome is a multisystemic disease with over 180 manifestations already described. In this report, the authors describe a patient presenting with a short stature, neurodevelopmental delay, and dysmorphisms, who had an episode of polyarticular arthritis at the age of three years and eight months, resulting in severe joint limitations, and was later diagnosed with 22q11.2 deletion syndrome. Investigation through Whole Genome Sequencing revealed that he had no pathogenic or likely-pathogenic variants in both alleles of the *MIF* gene or in genes associated with monogenic arthritis (*LACC1*, *LPIN2*, *MAFB*, *NFIL3*, *NOD2*, *PRG4*, *PRF1*, *STX11*, *TNFAIP3*, *TRHR*, *UNC13DI*). However, the patient presented 41 risk polymorphisms for juvenile idiopathic arthritis. Thus, in the present case, arthritis seems coincidental to 22q11.2 deletion syndrome, probably caused by a multifactorial etiology. The association of the *MIF* gene in individuals previously described with juvenile idiopathic arthritis and 22q11.2 deletion seems unlikely since it is located in the distal and less-frequently deleted region of 22q11.2 deletion syndrome.

## 1. Introduction

Juvenile idiopathic arthritis (JIA) is a heterogeneous group of diseases characterized by an arthritis of unknown origin affecting children, and is the most common chronic rheumatic disease of childhood and an important cause of disability. JIA is an umbrella term for all arthritides with onsets before 16 years, lasting longer than six weeks, and it is highly heterogeneous in terms of its etiology and clinical presentation. Its incidence ranges from 1.6 to 23 per 100,000 children annually and the prevalence of JIA is about 3.8–400 per 100,000 children in Europe, with a higher incidence in girls [1,2]. In the State of São Paulo, Brazil, the incidence has been estimated to be 1/2.880 (or 0.34/1000) [3].

The cause of JIA is not well known but involves the inflammation of the synovium and the destruction of tissues in joints, which can cause the early onset of oligo articular JIA. It is challenging to diagnose in some children who initially complain of pain and joint swelling, as there is no single blood test that can confirm its diagnosis [4].

Some monogenic forms of JIA have been described, linked to variants in the genes encoding perforin (*PRF1*), Munc13-14 (*UNC13D*), syntaxin 11 (*STX11*), tumor necrosis factor-α-induced protein 3 (*TNFAIP3*) [5], laccase (multicopper reductase) domain-containing protein 1 (*LACC1*), and nuclear factor interleukin 3-regulated (*NFIL3*) [1], besides syndromic JIA conditions such as Blau syndrome (*NOD2*), Camptodactyly-arthropathy-coxa vara-pericarditis (CACP) syndrome (*PRG4* and *TRHR*), multicentric carpotarsal osteolysis (MCTO) syndrome (*MAFB*) [2], and Majeed syndrome (*LPIN2*) [6].

However, as with most immune-mediated diseases, JIA is a rather multifactorial disease caused by undefined environmental agents overlapping a complex genetic background [7]. Twin and family studies have provided evidence for genetic contributions to JIA susceptibility, including monozygotic twin concordance rates ranging from 25% to 40% and the prevalence among siblings being 15 to 30 times greater than that of the general population [5].

Many HLA alleles, including HLA class I and class II genes, and common variants in non-HLA genetic loci have been identified in several works, including genome-wide association studies in different JIA subtypes [7,8,9,10,11,12,13,14].

22q11.2 deletion syndrome (22q11.2DS, MIM #188400 and #192430) is a multisystemic disease caused by a heterozygous microdeletion in locus 22q11.2, usually arising from a de novo event or being inherited following an autosomal dominant pattern. It is the most common microdeletion in humans, with an estimated frequency of around 1:4000 living births. This deletion arises from an unequal meiotic exchange involving four blocks of low copy repeats (LCRs), resulting in the loss of a 3-Mb interval (the typical deletion, occurring in over 90% of cases) or a proximal 1.5 Mb deletion (occurring in around 10% of cases); the typically deleted region contains approximately 45 functional genes [15,16,17].

A series of abnormalities occur in 22q11.2DS, especially those that are cardiovascular, palatal, endocrine, neuropsychiatric, immunological, and dysmorphic. However, any organ can be affected since a broad spectrum of different individual characteristics exists with extremely diverse clinical presentations, with over 180 manifestations already described [15,16,17]. Despite this, 22q11.2DS is under-recognized, and its diagnosis is delayed, especially if cardiac malformations are absent [18].

The occurrence of juvenile rheumatoid arthritis (JRA) in association with 22q11.2DS was first reported in two unrelated girls presenting with polyarticular joint pain at the ages of seven and five years, respectively; the authors questioned if there was a coincidental or an unusual complication of this condition [19,20]. In the following years, other anecdotal cases or small series with similar features were reported [21,22,23,24]. An attempt to justify this association was later proposed by the location of the macrophage inhibitory factor gene (*MIF*) at 22q11.23 [25], whose activity can be detected in the synovia of patients with inflammation and plays a role in the susceptibility to systemic arthritis. However, the precise location of the *MIF* gene was later established and is outside the typical proximal deletion responsible for most cases of 22q11.2DS.

In this report, the authors describe a patient presenting with a short stature, neurodevelopmental delay, and dysmorphisms, who had an episode of polyarticular arthritis at the age of three years and eight months, resulting in severe joint limitations. He was later investigated through Whole Genome Sequencing (WGS) and diagnosed with 22q11.2DS.

## 2. Patient and Methods

The patient was seen in the Clinical Genetics Service of the Teaching Hospital at the University of Campinas during regular outpatient consultations from 1998 to 2024.

WGS was performed using DNA extracted from peripheral blood in an Illumina platform after mechanical fragmentation and a PCR-free protocol. Data were processed to detect point mutations, copy number variants, and structural variants as per the best practices for the bioinformatics pipeline [26]. Quality metrics were a minimum coverage over 20× and an at least 90% depth above 15×. The reference genome was GRCh38/hg38. Variant nomenclature and classification followed the ACMG recommendations [27,28].

Sequence variants for target genes and single nucleotide polymorphism (SNP) previously associated with JIA were analyzed using the Franklin by Genoox (https://franklin.genoox.com, accessed on 5 February 2024) community platform [29] version 2024.1 from January 2024. For every variant of interest, the automated classification generated by Artificial Intelligence was checked and updated manually.

## 3. Results

The proband was the second child of a healthy, young, and non-consanguineous couple of Italian and Portuguese origin; an elder sister born at term died 5 h after birth due to pulmonary immaturity (*sic*). He was first seen at the age of two years due to a neurodevelopmental delay associated with dysmorphic features.

Pregnancy was uncomplicated, but delivery was at 34 weeks due to functional dystocia, with the baby measuring 46.5 cm (+0.71 SD) and having a weight of 2650 g (+0.97 SD), and being discharged home on the fourth day of life after a diagnosis of a heart murmur with spontaneous resolution in the following weeks.

Neuropsychomotor development was globally delayed, with almost absent speech and unintelligible words; he had severe learning disabilities and attended a school for children with special needs. At the age of 10, he began agitation and aggressive behavior, and at age 11, he presented with epilepsy. He currently (age 29 years) receives polypharmacological management, including chlorpromazine, clomipramine, valproic acid, risperidone, quetiapine, and diazepam, in addition to vitamin D3 due to osteoporosis and levothyroxine due to hypothyroidism.

An initial evaluation revealed the following dysmorphisms (Figure 1): a prominent metopic ridge, mild hooded eyes, a dysmorphic, folded, and anteverted left ear, a short nose with anteverted nares, long fingers, clinodactyly of the 4th and 5th toes, and hypotonia. His light iris color and pale skin and hair follow the familial pattern. At the age of 22 years, stature was 151 cm (−3.5 SD), with a weight of 40.4 kg (−4.07 SD), and an OFC of 54 cm (−1.79 SD).

The complementary investigation included routine karyotyping, screening for inborn errors of metabolism, serum CPK levels, an EEG, an audiological evaluation, an echocardiogram, and an abdominal ultrasound screening for visceral malformations, all within normal ranges. A brain CT scan at the age of four years revealed frontal subcortical atrophy. A skeletal survey at the age of six years detected Wormian bones and the posterior fusion of the 2nd cervical vertebra, the axis, and the 3rd cervical vertebra (C2–C3). Fiberoptic laryngoscopy at the age of nine years showed palatine insufficiency.

At the age of three years and eight months, he presented with a sudden episode of cyanosis in all four extremities, followed by edema and pain, which evolved into articular restriction in the metacarpophalangeal and interphalangeal joints of the hands and feet, in addition to the knees. A posterior complementary investigation two and a half years after the episode included an erythrocyte sedimentation rate (ESR) of 10 mm in the first hour, rheumatoid factor (RF) < 20 IU/mL, C-reactive protein < 0.01, antinuclear body (ANA) negative, and HLA haplotype DR3/DR52. The radiologic evaluation at that time showed an irregular metaphyseal border of the distal femora and periarticular osteopenia in the hands and feet.

There were no other similar episodes, but he developed functional limitations of the hands, feet (Figure 2), knees, and had hip movements progressing to an abnormal gait. A physical articular examination at adult age showed a 45° flexion deformity of the wrist at the left hand with a subluxation of the 1st and 2nd metacarpophalangeal joints and a deformity of the 3rd metacarpophalangeal joint; on the right hand, a 45° flexion deformity of the wrist with a subluxation of the 1st metacarpophalangeal and 1st distal interphalangeal joint; there was feet positional deformity and a reduced joint range of motion of the hip and knees.

A skeletal survey at the age of 19 years showed the following findings: an overall dysplastic aspect with a thinning of the long bones and diminished bone density, a vertebral posterior fusion of C2–C3, an ankylosis of the wrist and carpal bones (Figure 3), and coxa valga; erosive osteoarthritis signs such as bone erosion were absent.

WGS detected a 2.2 Mb heterozygous deletion at 22q11.21 (genomic position: chr22:18948676-21110520; GRCh38—hg38). This deletion encompasses 44 protein-coding genes, including 38 registered in the OMIM database, and is classified as pathogenic according to the ACMG criteria for CNV classifications (1A, 2A, 3C, and 5G). The deletion overlaps the proximal region between LCRs A and D, which is the most common region deleted in 22q11.2DS (Figure 4).

Other genes of interest for monogenic forms of JIA were analyzed, including *LACC1*, *LPIN2*, *MAFB*, *MIF*, *NFIL3*, *NOD2*, *PRG4*, *PRF1*, *STX11*, *TNFAIP3*, *TRHR*, and *UNC13D*; no pathogenic or likely pathogenic variants were found in this patient.

In addition, 41 SNPs, previously described as risk polymorphisms for JIA, were found and described in Table 1.

## 4. Discussion

JIA has been classified into several subtypes, including systemic, oligoarticular, and polyarticular JIA, psoriatic arthritis, and enthesitis-related JIA, with its occasional subdivision into negative and positive RF forms [1,7,30]. The present patient was clinically defined as having polyarticular JIA, RF-negative.

The clinical and laboratory features of similar polyarticular presentations in individuals with 22q11.2DS were previously called “JRA-like” [21] due to the differences from the common JRA: none developed uveitis, almost half of them were ANA-positive, and only a few were RF-positive [31].

In a review of 80 patients enrolled in a 22q11.2DS cohort, Sullivan et al. [32] found three with similar polyarticular features, establishing a frequency of 3.75%, 50 to 150 greater than in the general population. However, it is important to note that this prevalence of arthritis was based on a single case series; therefore, its real frequency remains unclear [33]. Among 150 patients with 22q11.2DS registered in the Brazilian Database on Craniofacial Anomalies (BDCA) of Brazil’s Craniofacial Project, none were reported as having JIA or JRA (unpublished data).

The *MIF* gene is one of the genes associated with an increased risk of susceptibility for JRA and is mapped on 22q11.23, between LCRs F and G, close to the 22q11.2DS region. This region is not encompassed in the deletion of the present patient and in most of the patients with 22q11.2DS, which presents a deletion of the proximal region between LCRs A and D (Figure 4).

The genetic predisposition to JIA is mainly due to HLA class II molecules (HLA-DRB1, HLA-DPB1), although HLA class I molecules and non-HLA genes have been implicated [7].

The major histocompatibility complex (MHC) region on chromosome 6 is packed with over 200 genes, playing an essential role in the immune system. Numerous associations between HLA polymorphisms and JIA subtypes have been reported in multiple populations. Within the HLA complex are genes encoding class I (HLA A, B, and C) and class II (HLA-DR, DP, and DQ) molecules. The class I allele HLA-A2 is associated with different JIA subtypes, especially in those with early-onset [5] Polyarticular RF-negative JIA which is associated with DRB1*08 and DPB1*03.

Once more, the present patient’s HLA haplotype is DR3/DR52, thus being part of the class II alleles that are less associated with JIA. However, there are no comprehensive genomic studies concerning this.

A systematic review of the literature suggests that about 100 different non-HLA candidate loci have been investigated for associations with JIA; independent confirmations are available for only a handful of candidate genes, including *PTPN22*, *MIF*, *SLC11A6*, *WISP3*, and *TNFA*. Differences in phenotype description can further complicate interpreting the different associations (or lack of associations) between genetic variants and phenotypes since JIA comprises several sub-phenotypes with distinct clinical features and outcomes [5].

The patient presented 41 risk polymorphisms for JIA, most in intronic and intergenic regions. Only two variants were detected in exonic regions (c.516C>A in the *LTBR* gene and c.1389T>C in the *PTH1R* gene), both resulting in synonymous protein changes. He also presented three polymorphisms in the *ANKRD55* gene and two in the *LTBR* gene, in addition to two in the intergenic regions *C5*<>*OT1*, *IL6*<>*TOMM7*, and *RUNX1*<>*LOC100506403*, which could suggest an additive effect for this loci. Sixteen of the 41 (39%) alleles present were homozygous, and again an additive effect was considered to be a potentially increased risk for JIA.

Genome-wide association studies have been used to investigate the polygenic basis of common complex disorders, such as cardiovascular disease, type 2 diabetes, breast and prostate cancer, and psychiatric disorders. Hundreds or even thousands of genetic variants/polymorphisms can be found in a single person, each having a small effect on the disease’s pathogenesis. However, a single variant is not informative for assessing disease risk; therefore, a sum of the risk alleles weighted by their individual effect size calculated based on their odds ratio for disease and prevalence in the population can be combined in a polygenic risk score (PRS) that captures part of the individual’s susceptibility to diseases. This theoretical and simplified model faces many practical challenges, such as the high cost of such tests, the heterogeneity of the clinical presentation in complex diseases, the lack of information on non-European ancestry populations, and the poorly understood role of environmental factors. Because a minority of the total genetic risk has been identified for most diseases, such scores typically have suboptimal performance. Thus, the clinical utility of PRSs has yet to be established, and there is a long road to go before they become useful tools for clinicians [34,35].

The PRS performs best for diseases of low prevalence (e.g., =1%) and high heritability, such as many rheumatic diseases [35,36]. Testing the PRS in general rheumatological settings can be challenging, as currently, no sensitive or specific tests are available to assist clinicians in making the diagnosis, sometimes leading to misdiagnosis [35,37].

Cánovas et al. [37] studied 11,958 individuals from three JIA case/control cohorts (UK, US-based, and Australia) with genome-wide SNP genotypes. The analysis considered the seven subtypes of JIA recognized by the International League of Associations of Rheumatology: systemic arthritis, oligoarthritis, RF-positive polyarthritis, RF-negative polyarthritis, enthesitis-related arthritis, psoriatic arthritis, and undifferentiated arthritis. In this study, the authors found a particularly high performance for enthesitis-related and oligoarthritis JIA, with the potential to augment current JIA subtypes’ diagnosis.

Considering this, although no PRS protocols are currently feasible for JIA, the growing knowledge in this field is promising in the near future.

## 5. Conclusions

In the present case, JIA seems coincidental to 22q11.2DS and was probably caused by a multifactorial etiology. The role of the *MIF* gene in similar previous cases is also questioned since this gene is located in the distal 22q11 region, between LCRs F and G, outside the most common region deleted in 22q11.2DS. Further reports of JIA in individuals with 22q11.2DS should be fully investigated to better understand this association’s frequency and real significance.

## Figures and Tables

**Figure 1 genes-15-00513-f001:**
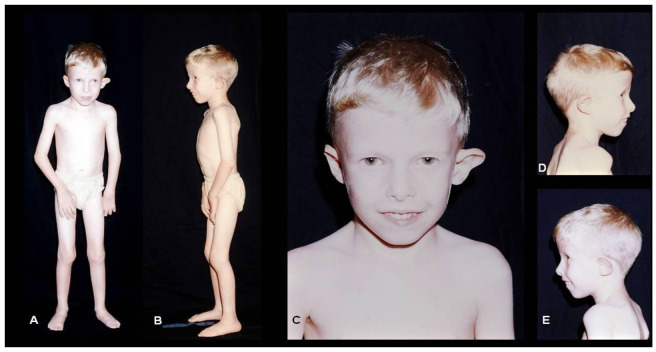
Photographs of the patient at the age of 4 years showing his overall body aspect (**A**,**B**) and facial features (**C**–**E**); note the dysmorphic left ear, hooded eyes, and pale skin and hair color.

**Figure 2 genes-15-00513-f002:**
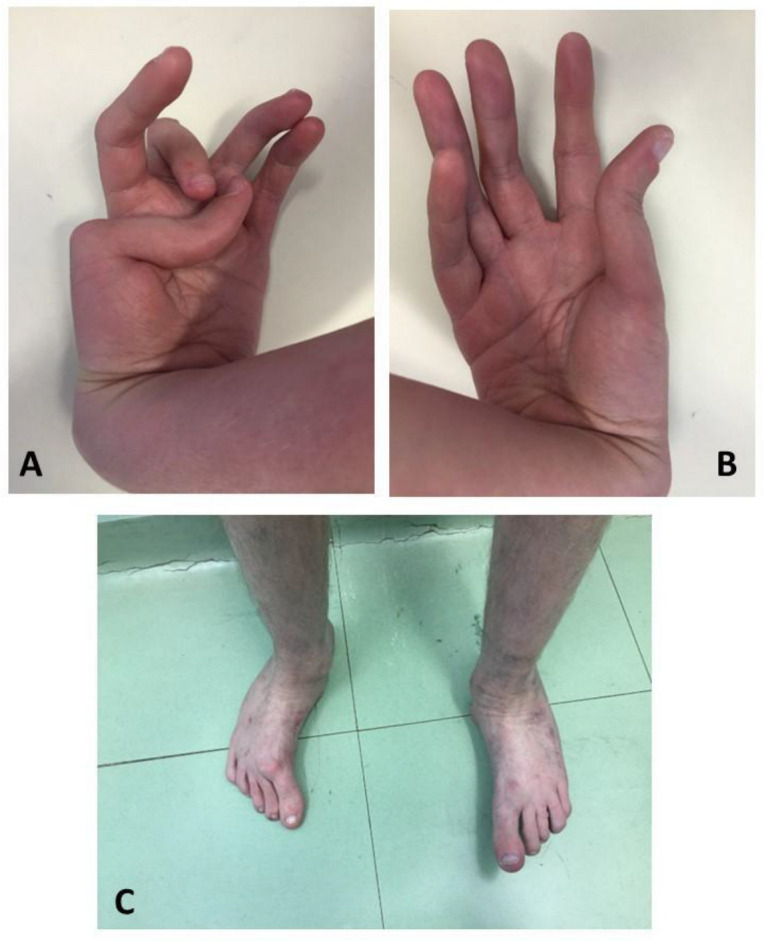
Articular deformities of the patient’s left hand (**A**), right hand (**B**), and feet (**C**) at the age of 23 years.

**Figure 3 genes-15-00513-f003:**
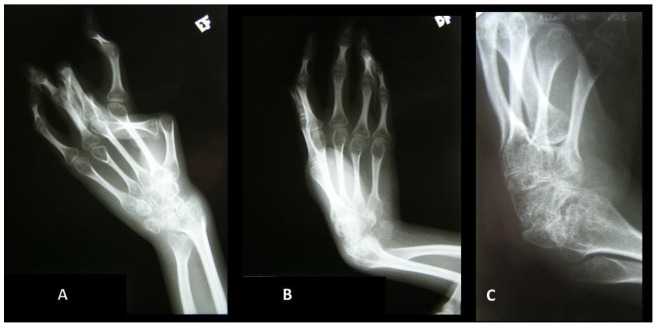
Radiographic changes in the left (**A**) and right hands (**B**), with a closer view of the carpal fusion in the right hand (**C**).

**Figure 4 genes-15-00513-f004:**
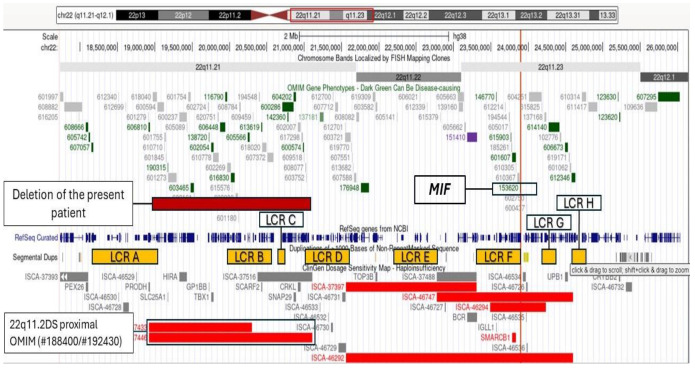
Schematic representation of the 22q11.2 locus; the purple bar shows the patient’s deletion in comparison to the proximal deletion region (the red bars below) and the distal deletion region (the remaining red bars). The location of the *MIF* gene is also emphasized. The orange boxes indicate the location of the low copy repeats (LCRs) A to H.

**Table 1 genes-15-00513-t001:** Risk polymorphisms for JIA detected in the patient.

Gene/Position	Polymorphism	cDNA/Nucleotide	Location	Zygosity	Ref.
*ADGRL2*	rs2066363	c.-101 + 10040C > T	intron 2	hom	[8,10]
*AFF3*<>*LINC01104*	rs6740838	n.100197037T > G	intergenic	het	[9]
*AHI1*	rs9321502	c.3166-11790G > T	intron 25	het	[9]
*ANKRD55*	rs10213692	c.484-2493A > G	intron 7	het	[1]
	rs7731626	c.484-4927C > T	intron 7	het	[1]
	rs71624119	c.484-974C > T	intron 7	het	[9]
*C5*<>*OT1*	rs10818488	n.1341T > C	intergenic	hom	[11]
	rs2900180	n.641-595A > G	intergenic	hom	[11]
*CLIC4*<>*RUNX3*	rs4648881	n.24870664G > A	intergenic	het	[9]
*COG6*	rs7993214	c.1827-11560T > C	intron 19 #	hom	[9]
*FAS*	rs7069750	c.31-410G > T	intron 2	hom	[1]
*HBP1*	rs111865019	c.-16 + 2616A > G	intron 1	het	[9]
*IL1B*	rs16944	c.-598T > C	UTR 5′	het	[11]
*IL2*<>*IL21*	rs1479924	n.122466445G > A	intergenic	het	[1,9]
*IL2RA*	rs706778	c.64 + 5102G > A	intron 1 #	hom	[1]
*IL2RB*	rs2284033	c.389-259C > T	intron 6	het	[1,9]
*IL6*<>*AS1*	rs1800795	n.54-321G > C	intergenic	hom	[12]
*IL6*<>*TOMM7*	rs7808122	n.22758461T > C	intergenic	hom	[9]
	rs6946509	n.22769871T > C	intergenic	hom	[1]
*IL19*	rs1800872	c.-149 + 1984T > G	intron 1	het	[12]
*IRF1*	rs4705862	c.*6424T > A	UTR 3′	het	[9]
*LNPEP*	rs27290	c.2219 + 553G > A	intron 12	het	[8,11]
	rs27293	c.2377-826A > G	intron 14 #	het	[1,11]
*LOC102723427*<>*CTN66*	rs727845	n.68142222A > G	intergenic	hom	[11]
*LOC105377621*	rs28362491	n.48 + 1438_48 + 1441del	intergenic #	het	[12]
*LTBR*	rs10849448	c.-174A > G	UTR 5′	het	[1]
	rs2364480	c.516C > A (p.A172=)	exon 5	het	[9]
*LURAP1L*	rs7042370	c.312 + 9047T > C	intron 1 #	hom	[13]
*NAA25*	rs17696736	c.1729-571T > C	intron 16	hom	[11]
*PADI4*	rs2240336	c.1048-34C > T	intron 10 #	hom	[12]
*PTH1R*	rs1138518	c.1389T > C (p.N463=)	exon 15	het	[9]
*PTPN2*	rs2847293	c.*3374T > A	UTR 3′	hom	[9]
*RNF215*	rs5753109	c.745-1498A > G	intron 6	het	[9]
*RUNX1*<>*LOC100506403*	rs812903	n.35340290T > A	intergenic #	het	[1]
	rs9979383	n.35343463C > T	intergenic #	het	[9]
*SGF29*	rs497523	c.-16 + 12513T > C	intron 1	het	[14]
*SNORA88*<>*WT1*	rs12795402	n.32234390T > C	intergenic	hom	[13]
*STAT4*	rs10174238	c.274-31983C > T	intron 4	hom	[1]
*TENM3*<>*DCTD*	rs7660520	n.182824168G > A	intergenic	het	[1,10]
*ZFP36L1*	rs3825568	c.76-801G > A	intron 2 #	het	[1,11]
	rs12434551	c.*2886T > A	UTR 3′	het	[9,11]

Key. het: heterozygous; hom: homozygous; UTR: untranslated region; #: Regulatory region.

## Data Availability

The data supporting this study’s findings are available from the corresponding author upon reasonable request.

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
