# Peer review of "Genome Sequencing in an Individual Presenting with 22q11.2 Deletion Syndrome and Juvenile Idiopathic Arthritis"

_genes, 2024, doi:10.3390/genes15040513_

Round 1
Reviewer 1 Report
Comments and Suggestions for Authors
The manuscript “Genome sequencing in an individual presenting with 22q11.2 Deletion Syndrome and Juvenile Idiopathic Arthritis” presents and describes a case of 22q11.2 deletion with associated JIA, with a genetic analysis of associated risk SNPs for JIA in the patient. The manuscript is well-written, and overall scientifically sound.
I have some minor key points I would like the authors to address:
· In the Introduction please rephrase the sentence “…and more than around common variant non-HLA genetic loci have been identified”, as it is not semantically correct.
· In the Introduction please correct the sentence “…is a multisystemic disease caused by a hemizygous microdeletion in the locus 22q11.2, usually arising from a de novo mutation.” – the deletion is heterozygous, and “does not arise from a de novo mutation”, it is usually (but not always) a de novo mutation/event
· Could you please expand on the elder sister that died after birth? What was the gestational age of the sister? Was she premature?
· Please indicate the age of the patient at which the photographs in Figure 1 were taken.
· Please add the points attributed to each of the criteria used for classification of the deletion, including the final score and category (pathogenic).
· Ensure consistency in human gene nomenclature (genes in all caps and italics).
· Could the authors discuss the evidence for potential usage of polygenic risk scores in the context of JIA, as multiple risk SNPs have been identified.
· It would also be interesting to describe which pharmacological treatments the patient underwent regarding the arthritis.
I hope the authors consider this criticism as constructive and hope it provides useful for the improvement of their work.
Author Response
In the Introduction please rephrase the sentence “…and more than around common variant non-HLA genetic loci have been identified”, as it is not semantically correct.
R: We rewrote the sentence.
In the Introduction please correct the sentence “…is a multisystemic disease caused by a hemizygous microdeletion in the locus 22q11.2, usually arising from a de novo mutation.” – the deletion is heterozygous, and “does not arise from a de novo mutation”, it is usually (but not always) a de novo mutation/event
R: We rewrote the sentence to clarify that it usually arises from a de novo event or is inherited following an autosomal dominant pattern.
Could you please expand on the elder sister that died after birth? What was the gestational age of the sister? Was she premature?
R: She was born at term, and we include this information in the text, but we don’t have further information regarding her.
Please indicate the age of the patient at which the photographs in Figure 1 were taken.
R: Done.
Please add the points attributed to each of the criteria used for classification of the deletion, including the final score and category (pathogenic).
R: The criteria used to classify the deletion as pathogenic are indicated in lines 171-172.
Ensure consistency in human gene nomenclature (genes in all caps and italics).
R: Corrected in lines 226-227.
Could the authors discuss the evidence for potential usage of polygenic risk scores in the context of JIA, as multiple risk SNPs have been identified.
R: We included a discussion on this issue in the last four paragraphs; there is currently no established PRS score for JIA in clinical practice.
It would also be interesting to describe which pharmacological treatments the patient underwent regarding the arthritis.
R: We don’t have this information, and currently, the patient receives no specific drug therapy for the arthritis.
I hope the authors consider this criticism as constructive and hope it provides useful for the improvement of their work.
R: Yes, we understand and thank the comments!
Reviewer 2 Report
Comments and Suggestions for Authors
The authors report the case of a child presenting with symptoms compatible with a diagnosis of JIA who was later found to bear a 2.2 Mb deletion in the 22q11 region. Since in the literature there are a few 22q11 deleted patients showing JIA features, the authors performed WGS in order to investigate whether the JIA is due to the deletion itself. They analyzed all the genes involved in monogenic JIA but were unable to find any mutations. Moreover, the MIF gene which might be responsible for the clinical picture is definitely located outside the deletion of this patient. By polymorphism analysis, the authors were able to identify several JIA-linked SNPs whose presence might suggest a multifactorial origin of the osteoarthritic symptoms. They conclude that in their case a causal association between the deletion and the JIA is unlikely.
Their genome analysis is sound and the conclusions are warranted, although the paper deals with a single patient.
In the legend of figure 1 the "ear" is written as "year".
In the legend of figure 4, last sentence, the term "detached" should be changed. The authors probably mean that it is outside of the deletion.
In the Discussion, last paragraph, the penultimate sentence starting from ANKRD55 is confuse. I suggest to clarify the point.
Comments on the Quality of English Language
Minor changes
Author Response
In the legend of figure 1 the "ear" is written as "year".
R: Corrected.
In the legend of figure 4, last sentence, the term "detached" should be changed. The authors probably mean that it is outside of the deletion.
R: We have changed for "emphasized".
In the Discussion, last paragraph, the penultimate sentence starting from ANKRD55 is confuse. I suggest to clarify the point.
R: We rewrote the sentence to clarify that, for some genes and intergenic regions, more than one polymorphism was found, which could suggest an additive effect in these loci.